# SRAM Compilation and Placement Co-Optimization for Memory Subsystems

**Biwei Liu**

College of Computer Science, National University of Defense Technology, Changsha 410073, China; liubiwei04@nudt.edu.cn

**Abstract:** Co-optimization for memory bank compilation and placement was suggested as a way to improve performance and power and reduce the size of a memory subsystem. First, a multi-configuration SRAM compiler was realized that could generate memory banks with different PPA by splitting or merging, upsizing or downsizing, threshold swapping, and aspect ratio deformation. Then, a timing margin estimation method was proposed for the memory bank based on placed positions. Through an exhaustive enumeration of various configuration parameters under the constraint of timing margins, the best SRAM memory compilation configuration was found. This method could be integrated into the existing physical design flow. The experimental results showed that this method achieved up to an 11.1% power reduction and a 7.6% critical path delay reduction compared with the traditional design method.

**Keywords:** SRAM; memory compilation; memory placement; co-optimization





## 1. Introduction

Modern microprocessors and various system-on-chip (SOC) technologies have large-capacity on-chip memory subsystems [1] composed of SRAM banks that are generated by a memory compiler. Multiple such memory banks are combined through glue logic to form cache, scratch-pad memory, shared buffers, or other on-chip memory subsystems. The capacity of on-chip memory subsystems continues to increase. Their area can reach 30–45% of the total chip area, and their power consumption is above 20% of the total chip power. At the same time, an on-chip memory subsystem is often involved in the critical timing path, which determines the full-chip frequency. Therefore, further improving the PPA of an on-chip memory subsystem is key to improving the PPA of the whole chip.

The present studies of on-chip memory subsystem optimization are carried out in different aspects. The first aspect is the optimization of memory banks. Some customized optimization designs have been proposed [2–6]. High-performance and low-power consumption memory compilation technologies have also been studied [7,8]. Efficient memory bank integration methods have been implemented. Gupta et al. proposed the use of heterogeneous memory banks to optimize the overall area of chips [9], while Yan et al. proposed custom optimization techniques [10]. The second aspect is the optimization of memory bank placement. Some evolution algorithms [11,12] or machine learning methods [13–15] demonstrated the automatic placement ability of memory banks. Cadence's mixed placer tool [16] is able to automatically place memory banks, which is claimed to get better performance and lower power consumption compared to manual placement. The third aspect is the optimization of glue logic. This is implemented by existing synthesis and place-and-route tools. Finally, FastMem [17] was proposed to optimize memory design at an architectural level.

However, the optimization aspects mentioned above are separated from each other in the existing design flow. Only the function of the memory bank is considered when compiling a memory bank. Only the geometric size of the memory bank is considered

when a memory bank is placed. Only the timing of the memory bank is considered when optimizing glue logic. IC designers have to reserve a large timing margin when a memory bank is compiled, resulting in the unnecessary area and power consumption.

To achieve an on-chip memory subsystem with high performance and low power consumption, this paper proposes a co-optimization method for memory bank compilation and placement. The location information of the memory bank is considered at the same time that the memory bank is compiled so that the memory bank compilation can be accurately constrained regarding timing. According to the timing constraints, the compiled memory bank's speed meets the requirements, and the power consumption is optimized.

## 2. Multi-Configured SRAM Compiler

The general design of the on-chip memory subsystem is shown in Figure 1. The main body is an SRAM bank array composed of SRAM banks. Outside the SRAM bank array is the glue logic, which includes a series of bus-merging, multiplexer, register operation, and bus protocol conversion logic. Finally, the memory system outputs to an external bus.

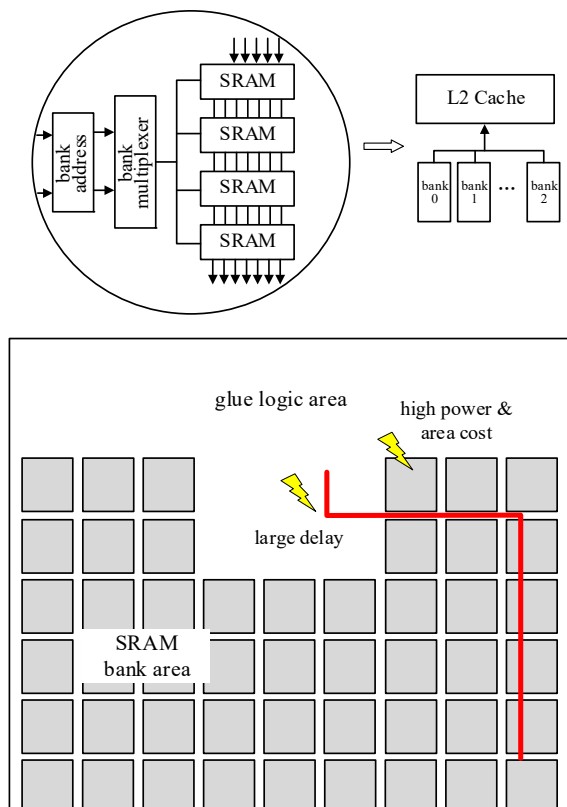

**Figure 1.** The structure of the on-chip cache.

Due to the larger size of the memory bank, the distance of the memory bank from the glue logic varied in different locations. For example, the memory bank in the corner of Figure 1 was far from the glue logic, so it was necessary to reserve a large timing margin and use a fast memory bank, as the red line in Figure 1. However, the memory bank in the center area was very close to the glue logic, and if the same compilation configuration was used as the memory bank in the corner, it led to waste in both power consumption and area. Therefore, using different compilation configurations for banks in different locations reduced area and power consumption, while increasing speed at the same time.

Based on an open-source memory compiler [7], we developed a multi-configuration SRAM compiler. The compilation configuration included three types: multiple thresholds, multiple sizes, and multiple aspect ratios.

### 2.1. Threshold Swapping

Threshold swapping is often used in integrated circuits to alternate between speed and power consumption. Low-threshold transistors are fast but consume large amounts of power, while high-threshold transistors are slow but consume less power and are more common in standard cells. The advantages of threshold replacement are that it occupies the same area, it does not need to adjust the floorplan of a chip, and it can replace insertion at any stage in physical design.

As shown in Figure 2b,c, the SRAM mainly included four parts: bit cell arrays, sensitive amplifications, inputs/outputs (IOs), and address decoders. Threshold replacement is only performed on the peripheral circuits, such as the address decoder and IO. We did not consider threshold swapping on the bit cell array because it could affect bit cell noise tolerance as well as other characteristics.

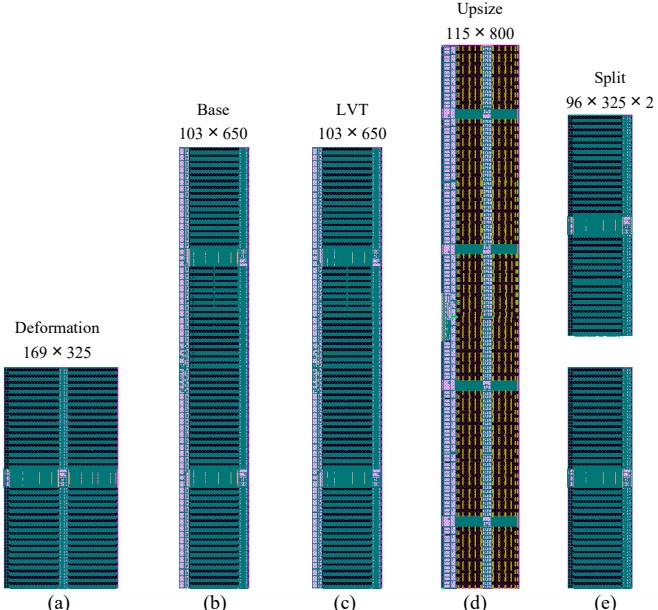

**Figure 2.** The multi-configuration of memory compilation.

### 2.2. Upsizing and Downsizing

Sizing is also a common way to alternate between speed and power consumption. Increasing size can increase speed, but it also brings increases in area and power consumption. Size adjustment takes a bit cell as the core, increases the size of the bit cell according to the performance requirements, and then adjusts the size of the peripheral circuits, such as the decoder, IO, and column multiplexer, according to the size of the bit cell. This approach significantly changes the bank area, which inevitably leads to changes in the floor plan.

### 2.3. Aspect Ratio Deformation

A memory bank can maintain the same capacity while its aspect ratio changes. One method is to change the column multiplexer in the memory body, for example, by halving the number of rows in the bit array but doubling the number of columns while adding a column multiplexer to select the output data, as shown in Figure 2a. Another way to keep the capacity the same is to change the depth and width of a bank, such as by halving the depth and doubling the width or vice versa. There are also other combinations of the above two methods.

In general, these methods keep both the capacity of a memory bank and the total area the same but significantly change the lengths of the word line and bit line so that the speed and power consumption change.

*2.4. Bank Splitting and Merging*

There are two ways to split an SRAM bank. One is to keep the depth unchanged and split the bit width. In this case, a small, divided memory bank can keep the same decoding circuit as the original large memory bank, and only the width of the bit cell array is reduced by half. The other is to keep the bit width unchanged and split the depth, as shown in Figure 2e. In this case, the height of a bit cell array is reduced by half, the address is reduced by 1 bit, and the decoding circuit is reduced accordingly. The two small memory banks maintain the original data bit width after splitting, which doubles the data lines. A multiplexer is required to select data externally from the small, multiple memory banks, which affects the logic design and wire routing.

Both methods shorten the word line or bit line, thus increasing the speed. However, after splitting, each unit requires peripheral circuits, such as decoding circuits, and the distance between the memory banks needs to be increased, so the area and power consumption both increase. Conversely, two (or more) memory banks can be combined into one, reducing area and power consumption while sacrificing performance.

## 3. Compilation and Placement Co-Optimization

This paper moved away from the original location-independent, homogeneous SRAM bank compilation method and proposed a bank-location-driven cooperative compilation process. For memory banks far from the glue logic, the delay was decreased by splitting, low threshold replacement, and upsizing. For memory banks close to the glue logic, area, and size were reduced by merging, high-threshold replacement, and downsizing. Power consumption was evaluated to achieve a balance of timing, power consumption, and area. The specific process is shown in Figure 3.

The design consisted of glue logic and several SRAM banks. First, based on the floorplan of the glue logic, the area of the glue logic was obtained, and the convergence point of the memory bank was selected as the reference point for the distance calculation of the memory bank. The convergence point could be the center of the glue logic area or a central location close to the port, and the glue logic area was set as the layout area on the floorplan.

Then, on the floorplan, the location of the unplaced area closest to the sink point was selected as the first bank layout location. The timing requirements for placing a bank at that location were calculated based on the distance from the sink point.

Then, we again exhaustively compiled possible memory banks in several dimensions, including depth, bit width, threshold, size, aspect ratio, etc., and selected the timing that met the requirements and had the least area per bit and the least power consumption per bit. Among these, timing requirements are described in detail in Section 3.1, and the exhaustive compilation process is described in detail in Section 3.2.

Finally, according to the size of the selected memory bank, we marked the layout area in the floor plan and recorded the total capacity of the layout. We continued to select the next closest location for bank compilation until the total size of the placed banks was as expected. This yielded the configuration and placement locations of all the banks.

After the above process, the memory bank could be split and merged, and the original glue logic may need to be fine-tuned. We reintegrated and synthesized the RTL code, performed placement and routing of the memory bank locations generated in the previous steps, and finally completed the entire design.

The processes of location-dependent setup, delay timing calculation, and exhaustive compilation are described in further detail below.

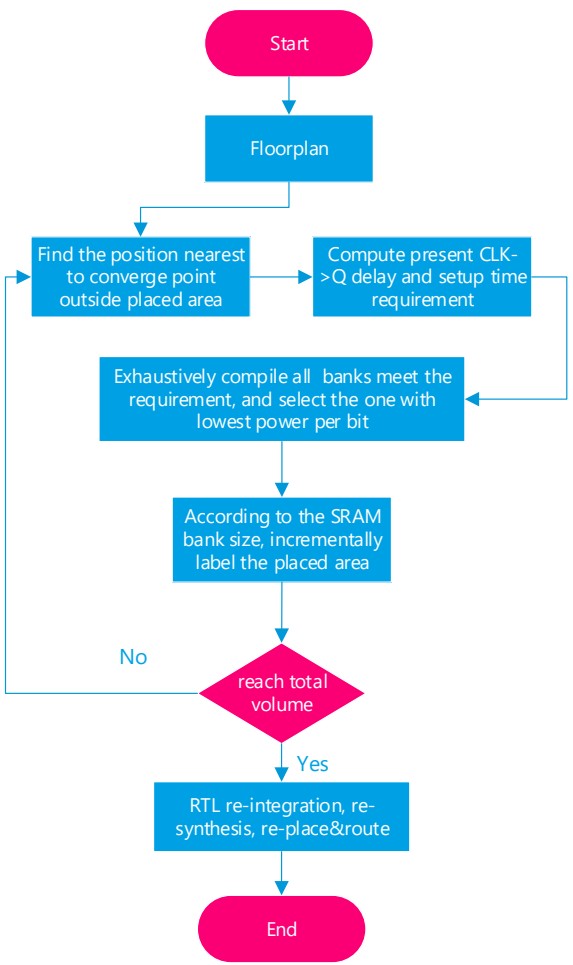

**Figure 3.** The compilation and placement co-optimization flow.

### 3.1. Determination of Position-Related Timing Constraints

In Figure 4, if the compilation and placement of memory bank 1 and memory bank 2 are completed and the layout position of memory bank 3 can be found in the unplaced area, its timing requirements can be divided into two items: (1) the setup time requirement from the glue logic to bank 3, and (2) the setup time requirement from bank 3 to the glue logic. This can be expressed as the following two inequalities:

$$t_{qm} + t_{gt} + t_d + t_{sr} + t_{mg} < t_p \tag{1}$$

$$t_{qr} + t_{gf} + t_d + t_{sm} + t_{mg} < t_p \tag{2}$$

where $t_{qm}$ and $t_{sm}$ are the delay from the memory bank clock to q and the settling time of the memory bank, respectively, which are the targets to be solved. $t_{gt}$ and $t_{gf}$ are respectively the delays of the combinational logic on the glue logic. $t_{qr}$ and $t_{sr}$ are the delay and setup time from the clock of the register on the glue logic to q, respectively, and they can be obtained by looking up the table in the timing library. $t_{mg}$ is a certain margin reserved to offset the influence of a small amount of wiring and crosstalk during actual wiring. $t_d$ is the delay caused by the distance from the sink to the memory bank, which can be calculated using the delay model related to the Manhattan distance [18].

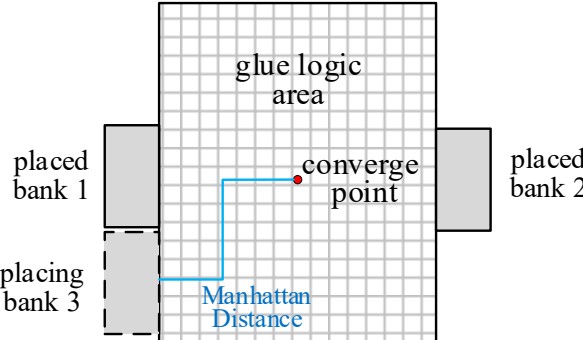

**Figure 4.** Co-optimization of memory compilation and placement.

The above two inequalities can be deformed to obtain the timing requirements of the memory bank:

$$t_{qm} < t_p - t_{gt} - t_d - t_{sr} - t_{mg} \tag{3}$$

$$t_{sm} < t_p - t_{qr} - t_{gf} - t_d - t_{mg} \tag{4}$$

It is also possible to consider using the design simplification constraint of effective clock skew (useful skew) so that the clock tree construction is more complicated. However, this does not affect the implementation of this method and is not discussed further in this paper.

### 3.2. Exhaustive Compilation of Memory Banks

The exhaustive memory bank compilation process also considered possible memory bank instance generation in several dimensions, such as depth, bit width, threshold, size, aspect ratio, etc. This was a large search space. To complete the search in a limited time, it was necessary to impose some limitations on these dimensions.

In terms of width and depth, they were required to be a power of 2, and the total capacity did not exceed 1 Mbit, which was convenient for the RTL code writing and physical implementation and did not skip obviously optimized configurations. In terms of threshold, three options were considered: high threshold, normal threshold, and low threshold. In terms of size, only two-bit cell sizes were available. More size options do not bring more optimization in actual engineering. In terms of multiple selectors, they had a power of 2, and the maximum did not exceed 16 because the speed of the memory was greatly reduced when it exceeded 16.

Under the above constraints, the compilation configuration of the memory could be reduced to 30~120 kinds, so that the exhaustive compilation could be completed in 3–4 h.

### 4. Experiments and Results

According to Section 3, an example was used to illustrate the process and demonstrate its effect. We used an on-chip shared buffer pool with a total capacity of 48 Mbit, a depth of 24,576, and a width of 256. It had an AXI bus protocol interface to interconnect other components on the chip. Based on a 28 nm technology, four implements were carried out for comparison: the traditional method, this paper method, Refs. [9,12] method. In all implements, the design was run to the post-route stage. Post-route netlists and spef files were extracted for timing and power analysis.

Figure 5 shows the design of the traditional method. It contained 96 homogeneous data memory banks (DMem). To realize the frequency of 500 MHz, the memory bank configuration options shown in the first row of Table 1 were selected. As shown in Table 2, the baseline design had an area of 4.75 × 4.2 = 19.95 um$^2$ and a critical path delay of 2.01 ns.

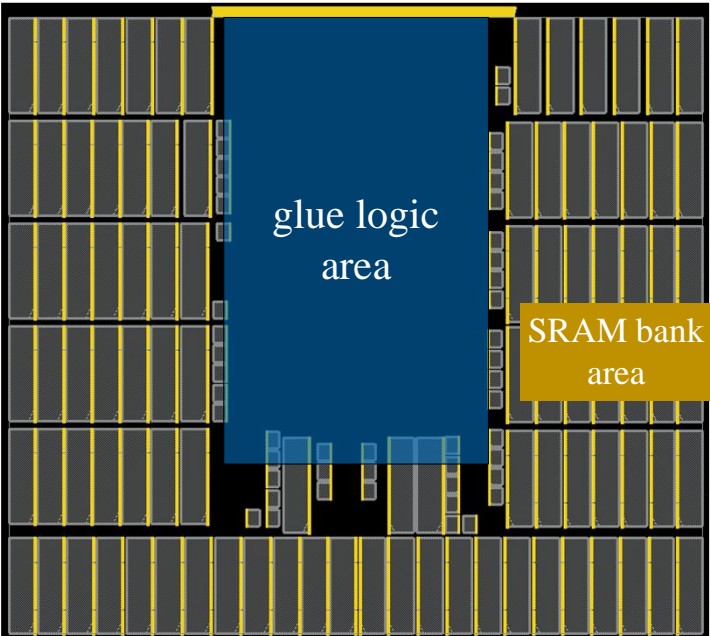

**Figure 5.** SRAM compilation and placement in traditional methodology.

**Table 1.** DMem bank performances under different configurations.

|  | $^{a}$ $t_{qm}$ (ns) | $^{a}$ $t_{sm}$ (ns) | Area (um$^2$) | $^{b}$ $P_{leakage}$ (mW) | $^{b}$ $P_{dyn}$ ($m$W) |
|---|---|---|---|---|---|
| DMem | 0.80 | 0.19 | 1 76 × 650 | 3.71 | 2.35 |
| DMemR | 0.85 | 0.20 | 330 × 325 | 3.71 | 2.35 |
| DMemRL | 0.68 | 0.18 | 330 × 3325 | 3.73 | 2.83 |
| DMemX2 | 0.97 | 0.19 | 330 × 3650 | 7.52 | 4.32 |
| DMemX2H | 1.25 | 0.29 | 330 × 3650 | 7.44 | 3.54 |

$^{a}$ Delay time is for WCL (worst-case low-temperature) corner. $^{b}$ Power is for ML (maximum leakage) corner.

**Table 2.** Design metrics and comparison with existing techniques.

|  | Area (mm$^2$) | Wire Length (mm) | Critical Path Delay |
|---|---|---|---|
| Traditional | 4.75 × 4.2 | 63.0 | 2.01 ns@WCL |
| This paper | 4.34 × 4.2 | 57.4 | 1.86 ns@WCL |
| Ref. [9] | 4.75 × 4.2 | 63.0 | 1.88 ns@WCL |
| Ref. [12] | 4.75 × 4.2 | 62.8 | 2.01 ns@WCL |

The configuration and placement of the memory bank shown in Figure 6 were obtained using the co-optimization method, and the selected convergence point was the upper area in the middle of the glue logic. It contained four different memory banks to realize the on-chip cache, and the specific parameters of each memory bank are shown in Table 1. DMemX2H was a memory bank whose depth was two times larger than that of the DMem and was replaced by high-voltage threshold transistors. The dynamic power consumption per bit of the memory bank at the end corner was reduced by 24.7% compared with the original memory bank, and the power consumption leakage per bit was basically the same as that of the reference design. DMemX2's depth was two times greater than that of the DMem. It was arranged on the periphery of DMemX2H, and its dynamic power consumption per bit was reduced by 8.1% compared with the original DMem. The capacity of DMemR was the same as that of DMem, but the depth was doubled and the width was reduced by half, which reduced the congestion of the wiring in the memory bank port as well as the power consumption. DMemR was placed on the outer periphery of DMemX2 to span a longer distance. DMemRL was a low-voltage threshold replacement based on

DMemR that was only used in the lower left and lower right corners to improve the speeds of these two key positions.

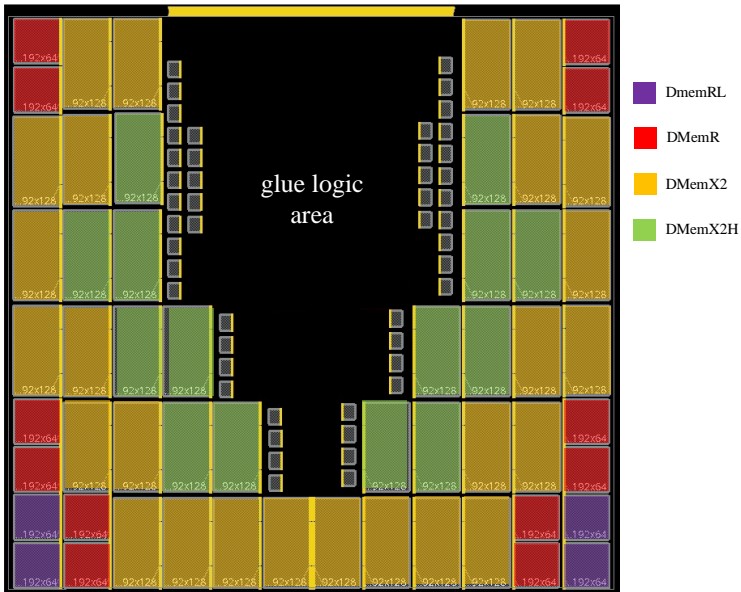

**Figure 6.** SRAM placement in co-operative methodology.

After trying to increase the size of the configuration, it was found that the unit power consumption was too large in this configuration, so it was not selected in all the positions.

A comparison of some key metrics of the design implemented by the traditional method, this paper method, and the other two state-of-the-art methods are shown in Tables 2 and 3. An industry-renowned STA tool is used for timing measurement. The analysis corner is set to the WCL (worst-case low-temperature) corner. In the WCL corner, the voltage is 0.81 V, the process is the worst case, and the temperature is $-40\,^\circ$C; the spef files are extracted as RCMAX.

**Table 3.** Instance count and comparison with existing technologies.

|  | Initial Inst Number | Post-Place Inst Number | Post-Route Inst Number | Post-Hold-Fix Inst Number |
|---|---|---|---|---|
| Traditional | 460 K | 900 K | 1.28 M | 1.44 M |
| This paper | 460 K | 880 K | 1.22 M | 1.31 M |
| Ref. [9] | 460 K | 900 K | 1.28 M | 1.44 M |
| Ref. [12] | 460 K | 900 K | 1.28 M | 1.44 M |

The area of this paper was reduced by 8.6% because many memory banks with doubled capacity were used, the gaps between the memory banks were eliminated, and the width of the whole chip was reduced. The critical path delay of this paper decreased by 7.5% due to the use of LVT banks in the corner positions. The area of Ref. [9] was kept the same as the traditional method because it only considered the num_height x num_width bank array, which limited further optimization. The critical path delay decreased by 6.5%, also due to the use of LVT banks in the corner positions. The area and critical path delay of Ref. [12] both stayed the same with the traditional method because it did not use other configurations of the memory bank and only optimized the placement of the memory bank, and in the traditional method, the placement is already fine-tuned manually.

Table 3 showed the number of instances of each method. In each design stage, the instance counts of this paper were significantly lower than the benchmark design. This is because the DMemX2 and DMemR memory banks were used to move the multiplexer into the memory bank, reducing the need for an external multiplexer and routing so that the

cell count was reduced by 9.0%. For similar reasons, the wire length was also reduced by 8.9%. Refs. [9,12] were not able to decrease the instance count because the memory bank configurations were nearly the same as with the traditional method.

Then, we compared the power consumption of this paper and other design methods. An industry-renowned power analysis tool is used for power measurement. The static power analysis method is used. The operation frequency was set at 500 MHz, the toggle rate on the data path was set to 0.2, the toggle rate on the clock path was set to 2, and the clock gating factor was set to 0.6. The analysis was carried out for the two corners, WCL and ML (max leakage). In the ML corner, the voltage is 0.99 V, the process is best case, and the temperature is 125 °C; the spef files are extracted as RCMAX. The results are shown in Table 4. The total power consumption under the optimized design for the WCL corner was reduced by 9.0%, and the leakage power consumption was basically the same as the baseline design. The total power consumption was reduced by 9.9% under ML, and the leakage power consumption was reduced by 5.8%. The total power and leakage power were both less than Refs. [9,12].

**Table 4.** Power consumption comparison.

|  | WCL Corner Power | | ML Corner Power | |
| --- | --- | --- | --- | --- |
|  | **Total** | **Leakage** | **Total** | **Leakage** |
| Traditional | 1.78 W | 0.24 W | 4.55 W | 1.54 W |
| This paper | 1.62 W | 0.23 W | 4.10 W | 1.45 W |
| Ref. [9] | 1.72 W | 0.24 W | 4.44 W | 1.49 W |
| Ref. [12] | 1.75 W | 0.24 W | 4.49 W | 1.51 W |

Finally, this co-optimization method was applied to different capacities (32~96 M bits) of memory subsystems to test its scalability. The results were shown in Table 5. The runtime increased with capacity, but even the largest design of switching buffer only needed 6.1 h to finish memory compilation and placement. For designs of different capacities, the critical path delay, power, and area of methods in this paper could all be improved relative to the traditional method.

**Table 5.** The scalability with memory capacity.

| Capacity | Runtime | Critical Path Delay | Power | Area |
| --- | --- | --- | --- | --- |
| 32 M bits | 3.7 h | 7.6% | 9.7% | 9.3% |
| 48 M bits | 4.3 h | 7.5% | 9.9% | 8.6% |
| 64 M bits | 4.9 h | 7.2% | 10.4% | 8.6% |
| 96 M bits | 6.1 h | 6.7% | 11.1% | 8.5% |

## 5. Conclusions

This paper proposes an on-chip cache design method with coordinated bank compilation and layout. Based on the different positions of memory banks on the chip, the method used various techniques such as splitting and merging, increasing and decreasing size, threshold replacement, and deformation to optimize the SRAM bank compilation instances, thereby improving performance and power consumption at the same time as reducing the area. The experimental results showed that this method achieved up to an 11.1% power reduction and a 7.6% critical path delay reduction compared with the traditional design method.

**Funding:** This research received no external funding.

**Data Availability Statement:** Data is unavailable due to privacy restrictions.

**Conflicts of Interest:** The authors declare no conflict of interest.

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
