# Peer review of "SRAM Compilation and Placement Co-Optimization for Memory Subsystems"

_electronics, doi:10.3390/electronics12061353_

Round 1
Reviewer 1 Report
There is no comparison with other optimization methods. The references are very limited, and need more recent sources.
Author Response
Point 1: There is no comparison with other optimization methods. The references are very limited, and need more recent sources.
Response 1:
1) In paper, we compared our method with the traditional methods in IC design industry (we call it baseline in Table 2 ~ Table 4). We change the name to traditional method in Table 2 ~ Table 4 to highlight the comparison.
2) We add 2 recent references
- J. Lin, Y. Deng , Y. Yang, J. Chen, Y. Chen, A Novel Macro Placement Approach based on Simulated Evolution Algorithm, IEEE/ACM International Conference on Computer-Aided Design (ICCAD), 2019
- J. Lin, Y. Deng, Y. Yang, J. Chen, P. Lu, Dataflow-Aware Macro Placement Based on Simulated Evolution Algorithm for Mixed-Size Designs, IEEE Transactions on Very Large Scale Integration (VLSI) Systems, 2021
3) We add comparisons of Ref. [7] and Ref. [12] in Table1~Table4. The results show our method is superior to other existing methods, because our method can explore the larger design space.
Reviewer 2 Report
The manuscripts present the co-optimization technique for SRAM compilation and placement. The contents are well described with examples. However, the technical details are too brief to catch the contents clearly.
For example, the reviewer would like to get the quantitative merits of this approach. As shown in the comparison table in Table 2, the speed performance gets improved while sacrificing the area. Does this intend to show the trade-off between optimized performance and any other aspects?
The reviewer does not have any other concerns on this paper except the above technical costs and potential challenges in this work.
Author Response
Point 1: Moderate English changes required
Response 1: The manuscript is checked by an English native speaker in MDPI editing service.
Point 2: The technical details are too brief to catch the contents clearly. For example, the reviewer would like to get the quantitative merits of this approach. As shown in the comparison table in Table 2, the speed performance gets improved while sacrificing the area. Does this intend to show the trade-off between optimized performance and any other aspects?
Response 2:
1) That’s not a tradeoff. In table 2, the three metrics of speed, area and wire length are all improved. And Table 3 shows the power is also improved. The only cost is it needs several hours runtime to compile suitable memory bank and place them; and the RTL code may become a little more complicate for instantiate different kinds of memory bank.
2) We add many details in the revised manuscripts :
We comparisons of Ref. [7] and Ref. [12] results in Table2~Table4. The results show our method is superior to other existing methods, because our method can explore the larger design space.
We also add the results applied to different capacity of memory subsystem to test its scalability. The results are shown in Table 5. The run-time increased with capacity, but even the largest design of switching buffer only needs to 6.1 hours to finish memory compile and placement. For designs of different capacity, the critical path delay, power and area of methods in this paper all can be improved relative to traditional method.
Reviewer 3 Report
In this work, the authors proposed performance improvement and optimization of the area of memory subsystem. This work is of particular interest for the research community working in this domain. The manuscript is well-written with detailed explanation of the results. However, there are some concerns of this reviewer to be addressed:
11. A comparison table is required to compare the proposed work with the available state of the art literature.
22. The authors mentioned that the experimental results show that this method is able to reduce the power consumption by about 9.9% and reduce the critical path delay by 7.5%. The authors are suggested to provide more details about the measurement setup.
33. The authors are advised to discuss the pros and cons of the compilation and placement co-optimization flow as shown in Fig. 3.
44. The manuscript should be reviewed by native speaker for language.
Author Response
Point 1: A comparison table is required to compare the proposed work with the available state of the art literature.
Response 1:
1) In paper, we compared our method with the mainstream methods in IC design industry (we call it baseline in Table 2~Table 4). We change the name to traditional method in Table1~Table 4 to highlight the comparison.
2) We add comparisons of Ref. [7] and Ref. [12] results in Table2~Table4. The results show our method is superior to other existing methods, because our method can explore the larger design space.
Point 2: The authors mentioned that the experimental results show that this method is able to reduce the power consumption by about 9.9% and reduce the critical path delay by 7.5%. The authors are suggested to provide more details about the measurement setup.
Response 2:
The details about timing and power measurement setup is as follow:
The design is implemented on a 28nm technology. In all cases, the design is run to post-route stage and post-route netlists and spef files are extracted. And then the post-route netlists and spef files, standard cell liberty files and memory cell liberty files are used for timing and power analysis.
An industry-renowned STA tool is used for timing measurement. The analysis corner is set to WCL (worst-case low-temperature) corner. In WCL corner, the voltage is 0.81V, the process is worst case, and the temperature is -40 ℃; the spef files are extracted as RCMAX.
An industry-renowned power analysis tool is used for power measurement. The analysis corners are set to WCL corner and ML (max leakage) corner. In ML corner, the voltage is 0.99V, the process is best case, and the temperature is 125 ℃; the spef files are extracted as RCMAX. The static power analysis method is used. The operation frequency was set at 500 MHz, the toggle rate on data path was set to a 0.2, the toggle rate on the clock path was set to 2, and the clock gating factor was set to 0.6.
Point 3: The authors are advised to discuss the pros and cons of the compilation and placement co-optimization flow as shown in Fig. 3
Response 3:
1) Positive side:Better area, performance and power consumption can be obtained at the same time.
2) Negative side:It needs some runtime to compile the optimized memory bank and try placement. And RTL code will become a little more complex to involve different memory bank.
Point 4: The manuscript should be reviewed by native speaker for language.
Response 4: The manuscript is checked by an English native speaker in MDPI editing service.
Reviewer 4 Report
May consider improving the grammar.
Mention about the scalability of the proposed system.
Author Response
Point 1: May consider improving the grammar.
Response 1: The manuscript is checked by an English native speaker in MDPI editing service.
Point 2: Mention about the scalability of the proposed system.
Response 2:
We also add the results applied to different capacity (32Mbit to 96Mbit) of memory subsystem to test its scalability. The results are shown in Table 5. The run-time increased with capacity, but even the largest design of switching buffer only needs to 6.1 hours to finish memory compile and placement. For designs of different capacity, the critical path delay, power and area of methods in this paper all can be improved relative to traditional method.
Round 2
Reviewer 1 Report
The references are increased but still a little bit limited. If possible, please increase more related references.
In addition, please briefly describe why the following spec is selected for experiment.
"a total capacity of 48 Mbit, a depth 204 of 24576, and a width of 256"
Author Response
Point 1: The references are increased but still a little bit limited. If possible, please increase more related references.
Response 1:
We search the related works again, and add 6 references as follows:
- Xingsi Xue,Aruru Sai Kumar,Osamah Ibrahim Khalaf,Rajendra Prasad Somineni,Ghaida Muttashar Abdulsahib,Anumala Sujith,Thanniru Dhanuja andMuddasani Venkata Sai Vinay, Design and Performance Analysis of 32 × 32 Memory Array SRAM for Low-Power Applications, Electronics 2023, 12(4), 834
- Sajib Barua, Umme Hani Irin, Md Minhajul Azmir, Md Maruf Abir Bappy Shayadul Alam, In 12nm FinFET Technology, performance analysis of low power 6T SRAM layout designs with two different topologies, 2022 IEEE 31st Microelectronics Design & Test Symposium (MDTS)
- Sriharsha Enjapuri, Deepesh Gujjar, Sandipan Sinha, Ramesh Halli, Manish Trivedi, A 5nm Wide Voltage Range Ultra High Density SRAM Design for L2/L3 Cache Applications, 2021 34th International Conference on VLSI Design and 2021 20th International Conference on Embedded Systems (VLSID)
- Xiang Gao, Yi-Min Jiang, Lixin Shao, Pedja Raspopovic, Menno E. Verbeek, Congestion and Timing Aware Macro Placement Using Machine Learning Predictions from Different Data Sources: Cross-design Model Applicability and the Discerning Ensemble, ISPD '22: Proceedings of the 2022 International Symposium on Physical Design, April 2022, pp 195–202
- Yi-Fang Chen, Chau-Chin Huang, Chien-Hsiung Chiou, Yao-Wen Chang, Routability-Driven Blockage-Aware Macro Placement, DAC '14: Proceedings of the 51st Annual Design Automation Conference, June 2014, pp 1–6
- Jai-Ming Lin,Szu-Ting Li,Yi-Ting Wang, Routability-driven Mixed-size Placement Prototyping Approach Considering Design Hierarchy and Indirect Connectivity Between Macros, DAC '19: Proceedings of the 56th Annual Design Automation Conference, June 2019, Article No.: 119, pp 1–6
Point 2: Please briefly describe why the following spec is selected for experiment. "a total capacity of 48 Mbit, a depth of 24576, and a width of 256"
Response 2:
Our method is suitable for L3 cache and shared memory in multi-cores processor. 48Mbit is a typical capacity of L3 cache or shared memory. The width of 256 is a typical width of AXI bus; therefore the depth is 48M / 256 = 24576.
However, we are not limited to this capacity. In Table 5, we expand to a range of 32~96Mbit. For other capacity in this range, our method achieved good results.